# Plastics and Micro/Nano-Plastics (MNPs) in the Environment: Occurrence, Impact, and Toxicity

**DOI:** 10.3390/ijerph20176667

**Published:** 2023-08-28

**Authors:** Edith Dube, Grace Emily Okuthe

**Affiliations:** Department of Biological & Environmental Sciences, Walter Sisulu University, Mthatha 5117, South Africa; gokuthe@wsu.ac.za

**Keywords:** plastic waste, degradation, microplastic, aging, internalization, toxicity

## Abstract

Plastics, due to their varied properties, find use in different sectors such as agriculture, packaging, pharmaceuticals, textiles, and construction, to mention a few. Excessive use of plastics results in a lot of plastic waste buildup. Poorly managed plastic waste (as shown by heaps of plastic waste on dumpsites, in free spaces, along roads, and in marine systems) and the plastic in landfills, are just a fraction of the plastic waste in the environment. A complete picture should include the micro and nano-plastics (MNPs) in the hydrosphere, biosphere, lithosphere, and atmosphere, as the current extreme weather conditions (which are effects of climate change), wear and tear, and other factors promote MNP formation. MNPs pose a threat to the environment more than their pristine counterparts. This review highlights the entry and occurrence of primary and secondary MNPs in the soil, water and air, together with their aging. Furthermore, the uptake and internalization, by plants, animals, and humans are discussed, together with their toxicity effects. Finally, the future perspective and conclusion are given. The material utilized in this work was acquired from published articles and the internet using keywords such as plastic waste, degradation, microplastic, aging, internalization, and toxicity.

## 1. Introduction

Plastics are synthetic organic polymers that differ in chemical composition and density such as polystyrene (PS), low-density and high-density polyethylene (LDPE and HDPE), polypropylene (PP), polyethylene terephthalate (PET), polyvinyl chloride (PVC), polyurethane (PU), polyester (PES), and polyamides (PA) [1,2], to mention a few. These can be classified based on the structure, method of polymerization, behavior on heating (thermoset/thermoplastic), physical, and mechanical properties (rigid, soft plastics, elastomers) [1,2,3]. These materials have a range of uses in building and construction, healthcare, sports and entertainment, electronics, agriculture, packaging, aeronautics, and others. The properties of plastics alone cannot allow for such a range of applications; however, the addition of additives such as fillers, stabilizers, pigments, foaming agents, lubricants, flame retardants, and plasticizers [1], among others, allow plastics to be designed for any desired application, with some designed to replace metal.

In 2021, global plastic production rose to more than 390 million metric tons [4] (up from 335 million metric tons in 2016), reflecting the steady and ever-increasing plastic demand. Though plastic products are cheap and easily produced, they take a long time to decompose and their waste is unmanageable worldwide, becoming an environmental threat [3]. Globally, 22% of plastic waste is either mismanaged or uncollected (as shown by heaps of plastic waste on dumpsites, in free spaces, along roads, and in marine systems), 49% is landfilled, about 19% is set on fire, and less than 9% is recycled [5]. Extensive utilization of single-use plastics and low recycling rates, together with poor waste management, are causes of plastic waste buildup in the environment [6,7]. This waste stream has significant adverse effects on the environment.

Uncollected plastic waste is reported to block drains and waterways, causing flooding and water stagnation and resulting in outbreaks of waterborne diseases [8]. On the other hand, plastics from landfills emit dangerous gases (methane and carbon dioxide) which contribute to climate change, while harmful chemicals leach into the soil and groundwater [9]. Incineration causes air pollution, with the released carbon dioxide contributing to global warming. Additionally, incomplete incineration of plastic often leads to the release of numerous toxins, including volatile organic compounds, polybrominated dibenzo-p-dioxins and furans, heavy metals, polycyclic aromatic hydrocarbons, polychlorinated biphenyls, and various other gaseous emissions into the environment [10,11]. These toxins have direct and severe health consequences, which include an increased risk of cancer and respiratory diseases, neurological disorders, damage to the immune and nervous systems, and cardiovascular diseases [12]. However, of additional concern is the release of microplastics (MPs, 100 nm–5 mm) and nanoplastics (NPs, <100 nm) to the environment. There is growing concern about the impact of these micro and nano-plastics (MNPs) on the environment and also the possible toxicity effects to living organisms, especially humans [13]. 

## 2. Entry and Occurrence of Micro and Nano-Plastics (MNPs) in the Environment

MNPs have been observed in water, sediments, soil, and air, in the form of fragments, fibers, pellets, foams, microbeads, sponges, rubber, or films. The polymer types of MNPs observed mostly include PE, PS, PP, PES, PET, PVC, PVA, and PA [14]. These can be categorized into either primary MNPs or secondary MNPs depending on their source. 

### 2.1. Primary MNPs and Secondary MNPs

#### 2.1.1. Primary MNPs

Primary MNPs are plastic particles less than 5 mm, released directly into the environment. The primary MNPs can be intentionally manufactured to either a micro or nano size by plastic manufacturing industries for specific commercial uses, from which they can find their way to the environment. During manufacture, MNPs can be tuned to specific properties such as viscosity, stability, physical appearance and specific abrasive effects, allowing them to be utilized as bulking agents or exfoliants, to prolong product shelf life, and for controlled release of active ingredients [15]. The ability of MNPs to encapsulate and slowly release materials has seen them be utilized in nanomedicine (for drug delivery) and in agriculture (for the release of nutrients, fertiliser, and other active ingredients). In nano-medicine, for instance, Dalela et al. synthesized poly(styrene-co-maleic acid)-paclitaxel nanoparticles (as a nano-drug delivery system) for the delivery of paclitaxel in solid tumors [16]. Meanwhile, in agriculture, Tian et al. coated urea with three polymers, that is, epoxy resin, vegetable oil-based PU, and liquefied starch-based PU for the slow release of urea [17]. 

Personal care products (cleansers, scrubs, makeup products, nail polish, and toothpaste) have been reported to have microbeads, which are responsible for the abrasive effects (including the exfoliating or cleansing effect) [18,19]. After utilization of these products, microbeads are ultimately washed off to wastewater treatment plants or directly to the environment. Unfortunately, due to their small size, some MNPs are not removed by the wastewater treatment processes; thus, they remain in water. Bashir et al. in their study estimated that personal care and cosmetic products in Macao city, China, may release into the environment more than 37 billion microbeads annually through wastewater treatment plants [19]. MNPs can also leak into the environment during their production, transportation, and storage.

MNPs can also be directly released to the environment through the abrasion of plastic materials during manufacture, usage, or repair, for example, wear and tear of rubber items (tires, rubber seals, footwear and others), wear and tear of synthetic textiles (especially during cleaning or washing), and peeling of coatings and paints [15]. A study by Hazlehurst et al. revealed that during domestic laundering in the United Kingdom, the microfibre released from textiles ranges from 6490 tonnes to 87,165 tonnes per annum, depending on fabric and laundering variables [20]. On the other hand, Paruta et al. reported paint peels to be the major source of primary MNPs in oceans [21].

#### 2.1.2. Secondary MNPs

Secondary MNPs are plastics less than 5 mm, derived from the breakdown and fragmentation of large plastic pieces on exposure to biological, physical, and chemical stressors [22]. The breakdown (degradation) mainly occurs naturally in the environment, through mechanical, hydrolytic, photolytic, oxidative, biological, and thermal degradation, Figure 1. 

Hydrolytic degradation involves the reaction of plastic polymer bonds with water molecules, resulting in the breakage of one or more polymer bonds forming shorter/smaller plastic segments [23]. This could be strongly contributing to the MNPs in aquatic environments. Tamayo-Belda et al. using polycaprolactone (PCL) showed that plastic can degrade through abiotic hydrolysis to NPs (PCL-NPs) [24]. 

In biodegradation of plastics, microorganisms secrete extracellular enzymes that attach to the plastic surface and subsequently, hydrolysis to short polymer intermediates takes place [25]. Schöpfer et al. demonstrated that MPs containing ester bonds are prone to enzymatic depolymerization through hydrolysis, as the large hydrolysable (poly(lactic acid)/poly(butylene co-adipate terephthalate) blend particles significantly mineralized and scanning electron microscopy revealed cracks on the surfaces, an indication of biodegradation initiation [26]. 

Oxidative degradation involves a reaction of a plastic polymer structure with oxygen, creating carbon–oxygen bonds that shorten the polymer chains. Photo degradation results from absorption of photons by plastics in the presence of a chemical oxidizer (such as air), subsequently leading to the breakage of polymer bonds. Thermal degradation arises from modifications in polymer properties due to increased temperature. Warm/ hot climates with a lot of UV light and the heat waves caused by climatic change accelerate photolysis, thermo-oxidative and photo oxidative degradation [23]. Meides et al. showed that polypropylene on exposure to accelerated weathering conditions (3200 h) of thermal and photochemical oxidation can degrade, with a 192 µm diameter particle forming 100,000 particles [27]. Egger et al. reported the fallout of plastic particles (with size between 500 µm and 5 cm) from the North Pacific Garbage Patch to the deep sea beneath, signifying degradation probably due to photochemical oxidation on surface waters [28].

Mechanical degradation arises from the fragmentation of plastics as a result of external forces in the environment. Wind or moving water or waves can result in the collision and abrasion of plastics with hard surfaces such as rocks and sand, resulting in fragmentation [15,29]. 

### 2.2. Entry and Occurrence of MNPs in the Soil, Water and Air

Both primary and secondary MNPs end up in the environment (soil, water, and air), from which most organisms are exposed. These have been reported in organisms (plants and animals including humans) and also in products such as salt, water, and beer, to mention a few [30]. Though MNPs at times occur in the soil and air, these are washed down to water bodies (Figure 2) and oceans (particularly the sea bed), which are the major sinks of these particles [31].

#### 2.2.1. Entry and Occurrence of MNPs in the Soil

The farming system is the major food production system especially in developing countries. The land and its soil quality are important resources, as the majority of the population in developing countries survives on subsistence farming [32]. Additionally, the soil provides important ecosystem functions to all living organisms. Unfortunately, the soil is now a reservoir for plastic waste. Plastic fragments enter the soil through many routes such as landfill dumping, atmospheric deposition, street runoff, and agricultural practices [33], among others, as illustrated in Figure 2.

i.MNPs from landfill dumping

Landfills are utilized to bury large amounts of waste, including plastic. They are the major depository of waste plastic, with 49% of the plastic waste generated globally deposited in these landfills [5]. The presence of microorganisms, moisture, air, heat, and other physical conditions allow plastics in landfills to degrade to secondary MNPs. Primary MNPs can also be dumped in landfills. Landfills are thus major sinks of MNPs in the soil. For example, Mahesh et al. reported the presence of MPs (ranging from 180 to 1120 MP particles/ kg of soil) in an open urban landfill site [34], while Afrin et al. observed MPs from LDPE, HDPE, and cellulose acetate (CA) in a Bangladesh landfill site [35]. These MNPs can be transported by agents such as air, water, or other organisms from landfills to new environments in the close vicinity. Soil invertebrates either push or ingest MNPs to deposit them in another environment. For example, earthworms have been reported to ingest MNPs on surface soils and excrete them in deeper soils [36]. Light MNPs are also blown by wind [37] or washed away by water runoff, ending up in remote areas, aquatic systems, or waste water treatment plants. 

ii.MNPs from sewage sludge

Wastewater treatment plants receive MNPs from industry, landfills, urban runoff, and domestic wastewater. About 90% of these MNPs are retained in the sludge [38,39,40,41]. Di Bella et al. revealed the presence of diverse plastics (fragments and fibers of PE, PET, PP, polybutadiene (PB), and PES) in a waste sludge [38]. Harley-Nyang et al. reported that from one wastewater treatment works in the UK, between 1.02 × 10^10^ and 1.61 × 10^10^ MNPs are retained in the sludge every month [42]. The sludge is either utilized as a fertilizer in farmlands or deposited in landfills, ultimately contaminating the soil with MNPs. Weber et al. observed MNPs down to 90 cm depth in farmlands, 34 years after the last sewage sludge application, with the highest concentrations being in regularly ploughed topsoils [43], showing the role being played by sewage sludge in introducing MNPs in soils. Unfortunately, the MNPs added to farmland soil through sludge as a biofertilizer have been reported to spread beyond the applied areas, resulting in the contamination of new areas [44].

iii.MNPs from agricultural practices

Agricultural practices contribute to the majority of MNPs in farming soils. Primary MNPs enter soils through the use of plastic-encapsulated slow-release fertilizers, pesticides, and seed coatings, as these remain in the soil after releasing the active ingredient [45,46]. Irrigation with MNP-contaminated wastewater and the use of waste sludge containing plastic fragments for compost also contribute to the MNPs in the soil. Work by van Schothorst et al. showed that compost samples from some municipal organic waste in Netherlands contained 2800 ± 616 MPs kg^−1^ [47]. In China, 2400 ± 358 MPs kg^−1^ (mostly fibers and films) were also reported in compost [48]. 

Other plastic materials utilized for different agricultural practices such as plastic mulch films, greenhouse films, plastic drip irrigation systems, protective nets, and plastic irrigation pipes, to mention a few, degrade to secondary MNPs in the soil [49]. For example, Li et al. revealed that after three decades of plastic mulching, topsoil (0–10 cm) accumulated MPs ranging between 7183 particles/kg and 10,586 particles/kg, with plastic mulching contributing 33–56% to the total MPs [50]. Topsoil from a Spanish farm that was mulching for more than 12 years was also reported to contain 2242 ± 984 MPs kg^−1^ [47].

iv.MNPs from other sources

Uncollected waste due to exposure to harsh environmental conditions also degrades to MNPs. Unfortunately, there are no studies on MNPs in the soil around illegal dumping or uncollected plastic waste. However, the soils in urban, industrial, and recreational areas have been reported to contain MNPs [51,52]. Though these could be from the degradation of different materials in these areas, degraded uncollected plastic waste also contributes to these MNPs. Burning of uncollected waste is very common. The bottom ash after incineration is reported to be a source of MPs [53].

#### 2.2.2. Entry and Occurrence of MNPs in the Air

The air is polluted with MNPs, especially around urban areas. MNPs have been reported in aerosols [54,55], dust [55], and atmospheric deposition [56]. Airborne MNPs settle on the land surface due to gravity or are washed down by rain. Pandey et al. revealed the presence of fragments, films, spherules, and fibers, of PP, PS, PE, PET, PES, and PVC, mostly  <1 mm in aerosols and street dust [55]. The wear and tear of plastic or polymeric materials, fiber dust from synthetic clothing, city dust, dust from poorly managed landfills, farm dust, synthetic rubber tire abrasion, plastic furniture deterioration, waste incineration, and emissions from synthetic polymer industries release MNPs into the air [57]. Industries that produce plastic and polymeric materials also release airborne MNPs during production. Sun et al. demonstrated the presence of airborne MPs in the air around a poly (ethylene:propylene: diene) rubber industry [58]. MNPs, because of their small size and light weight, can be carried over long distances by the wind. Their presence in very remote areas confirms their long-distance transportation [57,59,60].

i.MNPs from the wear and tear of plastic or polymeric materials

Mechanical abrasion from contact between the plastic or polymeric materials and surfaces results in wear and tear. For example, vehicles tires which are made from synthetic polymers (rubber) release wear particles through mechanical abrasion [61]. Tire wear particles are regarded as microplastics due to their size, their physio-chemical properties, and the presence of synthetic polymers in the tire composition [62]. The particles have been observed in road dust, in the air, and in nearby water runoff. These particles can be transported through air to remote areas, or they can be washed away by water runoff to either water treatment plants or to natural water systems. Goßmann et al. analysed the dust trapped in urban spider webs and reported the presence of car tire wear, together with clusters of PET and PVC, confirming the presence of MPs in urban air [63]. Sieber et al., utilizing dynamic probabilistic material flow analysis, concluded that 219 ± 22 ktonnes of rubber particles accumulated in Switzerland’s environment since 1988 [64], confirming the contribution of rubber particles to the MNPs in the environment.

ii.MNPs from synthetic textiles and clothing

Synthetic textiles and clothing release synthetic fibers, such as PES, nylon (PA), PP, acrylic, and spandex (from polyether and polyurea) during tumble drying when laundering. O’Brien et al. showed that drying a 660 g blanket for 20 min in a domestic dryer releases about 1.6 to 1.8 fibres/m^3^ into the surrounding air, with the inbuilt filtration capturing ≈ 1.1 ± 0.3 × 10^6^ fibres [65]. Spinning, weaving, and clothes-making processes also release fibers to the air [66,67]. Nail and hair salons release MNPs into the environment (air, soil, and water). Wigs and synthetic hair fibers are made of polymeric materials (acrylic, PES, and nylon, to mention a few) and thus release MNPs during combing, brushing, and cutting. Chen et al. reported the presence of acrylic, rubber, and PU fragments (<50 μm) in the air from nail salons [68], showing that acrylic nails and nail polish also contribute to the MNPs released by salons to the environment.

iii.MNPs from the sea spray

MNPs, due to their light weight and small size, are always floating on the ocean surface. A combination of sea spray, waves, and wind has been shown to create air bubbles that contain MNPs in the water and these release MNPs into the atmosphere when the bubbles burst [69]. MNPs have been observed in the atmosphere around the ocean. For example, Caracci et al. reported the presence of mostly PE, PP, polyisoprene (PI), and PS in Atlantic ocean atmosphere [54]. The emission of MNPs through sea spray (Figure 2) is reported to be influenced by factors such as particle size, density, and concentration. Harb et al. observed that aerosolization increased with an increase in the concentration of MNPs in water and also with a decrease in particle size [70]. Yang et al. reported levels of MNPs emitted from sea spray to be up to 24 quintillion pieces or 773 tons annually for particle sizes between 0.3 µm and 70 µm [71], showing that oceans are a significant source of MNPs in the atmosphere. Strong ocean winds can transport these MNPs further away to more remote areas in oceans and terrestrial areas [72]. A recent study by Preston et al. demonstrated that flat fibers are efficiently transported over long distances compared to spherical counterparts, which also suggested that oceans could be major sources of MNPs in the atmospheric [73]. 

iv.MNPs from wind abrasion and erosion

Wind abrasion occurs when rough particles blown by wind break down plastic by rubbing on its surface. Agricultural plastic film that has been exposed to UV radiation has been shown to easily break into debris (ultimately forming MNPs) when exposed to wind abrasion [74]. Bullard et al. demonstrated that during wind abrasion (aeolian saltation), MPs can be fragmented, as shown by the reduction in the diameter and weight of the spherical MP beads [75].

On the other hand, wind erosion occurs when loose particles including MNPs are moved from one location to another. Wind erosion has been shown to facilitate the transport of MNPs from the soil including landfills to the atmosphere and aquatic ecosystems [76]. Rezaei et al. demonstrated the role of wind erosion in spreading MPs in terrestrial environments as wind-eroded sediments contained 20.27 mg kg^−1^ of light density MP [77]. Soil erodibility, wind speed, and MNP type have been shown to influence the erosion of MNPs from the soil surface to the atmosphere [78,79]. 

Air pollution is one of the major human health threats, resulting in many deaths globally [80]. Recently, MNP particles were identified as emerging airborne anthropogenic pollutants [56,81] whose atmospheric accumulation and further deposition should be monitored. MNP aerosols or dust could be a significant pathway for inhalation by living organisms including animals and human beings (Figure 3), resulting in toxicity effects.

#### 2.2.3. Entry and Occurrence of MNPs in Water Systems

MNPs enter water systems using varied routes, such as urban runoff, atmospheric deposition, waste leakages (from sewage and poorly managed landfill effluents), wastewater treatment plants, and inappropriate waste disposal [82,83]. For example, Apetogbor et al. reported the presence of MPs in water and sediment samples from Plankenburg River, South Africa [84]. Khan et al. also observed MPs in sediments, water, and fish (*Schizothorax plagiostomus*) samples, from Swat River, Pakistan [85]. Most low-income countries lack adequate waste collection and disposal services [86], and waste is dumped arbitrarily, including COVID-19 pandemic personal protective equipment [87,88]. Morgana et al. demonstrated the potential of single-use face masks (on degradation) to release MNPs into water systems [89]. Wang et al. proved that the middle layer of masks is sensitive to UV weathering with a single weathered mask releasing more than 1.5 million MNPs to the water systems. They also demonstrated that sand in shorelines can aggravate the release of MNPs from masks using physical abrasion, with more than 16 million particles released from just one weathered mask [90]. 

i.MNPs from water runoff

The water runoff carries large amounts of MNPs to aquatic systems. These MNPs are washed away from road dust, atmospheric deposition, abrasion of road markings, degrading plastics in open dumps, peels from paint coatings, farm soils (due to agricultural practices that release MNPs as mentioned in Section 2.2.1), landfill leachates, and industrial waste [91]. Beni et al. reported higher concentrations of MPs in the runoff from croplands utilizing municipality sludge compared to croplands utilizing cattle manure as sources of fertilizer. Fibers and fragments were the most detected; however, films, foam, and beads were also observed [92]. Ross et al. estimated that separate urban runoff outlets released between 1.9 million to 9.6 billion MPs every rain event to receiving water bodies [93], confirming that water runoff is a major pathway for transporting MNPs from land to aquatic environments (Figure 2).

ii.MNPs from industrial and municipal waste

Industrial waste (especially from polymer and textile processing) as well as municipal waste (especially sewage) either leaks or is discharged to water streams or to the sea [94,95]. Chan et al. showed that industrial wastewater from a textile processing mill in China contained MNPs in the form of fibers (361.6 ± 24.5 MNPs L^−1^) [96]. Discharged wastewater systems of plastic production plants in Germany were shown to contain MPs, mostly consisting of PE, PET, PP, and PVC. [97]. Similarly, Bitter et al. reported the presence of mostly PE, PP, PA, and PET in industrial waste [98]. 

Municipal waste is also a source of MNPs in aquatic systems. For example, in the UK, the discharge of raw sewage to some streams has been reported [95,99], while in South Africa, Kretzmann reported the seepage of sewage into the Vaal river [100]. This releases MNPs directly to aquatic environments. Dharmaraj et al. observed coliforms associated with fecal matter on MP from the Adyar river [101], showing how sewage leaks release MNPs into water bodies [102].

iii.MNPs from fishing activities

Current fishing gear consists of plastics such as PE, PP, and PA, to mention a few. The wear and tear of fishing gear during fishing activities releases MNPs to aquatic systems. For example, the Danish seine ropes can be dragged along the seabed for a long distance, with frictional force causing the ropes to wear. Syversen et al. showed that between 77 and 97 tons of plastics are added to the sea annually in Norway from wear and tear during Danish seining [103]. The Norwegian fisheries in general are estimated to release about 208 tons of MNPs per annum [104]. 

Fishing gear also gets lost and abandoned in aquatic environments due to different causes such as vandalism, bad weather, and gear malfunctions. When it degrades, MPs are released to these aquatic environments and these MNPs have been detected in rocks and sediments in areas with ghost fishing gear [105]. Kuczensk et al. estimated that about 45,000 tons of plastic pollution is released annually from lost fishing gear [106]. The level of MPs released is dependent on the fishing litter density and the environmental conditions. Wright et al. reported that lost or discarded fishing gear on beaches of the English Southwest Peninsula, Great Britain (with high fishing litter density), could release about 1277 ± 431 MP pieces m^−1^ [107]. 

iv.MNPs from garbage patches

Garbage and debris accumulate and float on oceans. This garbage consists of mostly plastic and fishing nets and gear [108], which on degradation contribute to MNPs in oceans. With the continuous entrance of plastic waste into oceans, the size of patches is expected to increase, together with the concentration of MNPs in oceans (due to degradation, especially in marine environments). The fallout of plastic particles from the North Pacific Garbage Patch to the deep sea beneath confirms the degradation of plastic from these plastic patches [28]. Zhao et al. showed the presence of MNPs throughout the water column of the eastern North Pacific Subtropical Gyre [109]. Different polymer types of MNPs have been observed, an indication that different polymer plastics find their way to the oceans. For example, Li et al. reported the presence of PET, PE, PA, rayon, and polyvinylidene chloride (PVDC) microplastics in the seawater of the southern Indian Ocean [110]. Coastal garbage patches are reported to have the same plastic concentrations as the Great Pacific Garbage Patch, with the mean mass within hotspots being about 5161 g km^–2^ [111]. Garbage patches thus contribute highly to the MNPs in oceans. The degradation of this plastic debris also releases other chemicals (from plastic additives). Fauvelle et al. reported a significant additive release from plastic at the ocean surface to the deep seawater [112]. 

### 2.3. Aging of Microplastics

MPs, after release from the parent material, further experience different aging processes, such as biodegradation, hydrolysis, thermal oxidation, chemical oxidation, photooxidation, and physical abrasion depending on the weathering conditions they are exposed to [113]. This has been shown to result in MPs that are embrittled, rough, flaky, and cracked. Additionally, surface properties are altered [114]. Once the surface properties (shape, size, structure and chemical composition) are altered, particles become smaller (until nanoplastics are formed), surfaces become rough, and surface functional groups are exposed, resulting in enhanced interactions with other compounds [115], including biomolecules and other pollutants. The aged MPs or resulting NPs are easily ingested (as illustrated in Figure 4), absorbed, internalized, and translocated within organisms, ultimately resulting in toxicity [116]. Toxicity can be from both the particles and the adsorbed pollutants [117,118,119].

When exposed to conditions such as solar radiation, MPs have also been shown to generate persistent free radicals on their surfaces [120,121,122], which are accompanied by reactive oxygen species (such as O_2_^•−^ and ^•^OH) [121] which can have adverse effects on organisms including humans. Aged MPs are more toxic than the pristine counterparts [123] and this has been attributed to the altered surface properties and the generated ROS [116].

## 3. Uptake and Internalisation of MNPs by Living Organisms

The uptake, internalisation, and translocation of MNPs are influenced by their particle size, chemical composition, shape, surface charge, and hydrophobicity. Small sizes and hydrophobicity of the MNPs enhance their translocation across the biological membranes, by direct passage through the porous structures of cell envelopes or through the cell wall due to enhanced cell membrane permeability during cell cycling and at times through endocytosis [124]. MNPs are taken up by organisms through exposure to MNP contaminated drinking water, air, soil, and diet (Figure 3).

### 3.1. MNP Uptake by Plants

Soils are reservoirs for MNPs. MNPs have been reported in plant parts including the roots, fruits, and vegetables, confirming their uptake and accumulation in plants [125,126]. MNPs are absorbed through the root hairs. NPs can be directly taken in by plants through the apoplastic pathway (through cell wall) due to their small size. MPs, on the other hand, cannot be directly absorbed by plant tissues due to their large size; however, these enter cells through cracks on plant lateral roots (crack entry) [127] and also through plant wounds. Recently, MPs were reported to promote the rounding of the apical epidermal cells, pulling apart the protective layer between the epidermal cells and forming holes which allow MPs into the roots [128]. MPs can also cause deformations on the plant cell wall and the deformations become another entry point. Dong et al. demonstrated that larger MPs (1219.7 nm) can enter the carrot roots and accumulate in the intercellular layer suggesting the deformation of cell walls by these MPs, and creating larger pores which allow further entry of larger MP particles [129]. Once in the plant roots, MNPs can be translocated from the roots to the shoots through the transpiration pull of the vascular system, which further promotes more MNP uptake by plants [127]. 

Foliar uptake has been shown to be a source of MNPs in terrestrial plants. MNPs are ubiquitous in the atmosphere due to airborne plastic pollution, and thus can be deposited to the plant aerial parts, particularly leaves. Guo et al. foliar exposed soil-grown maize and soybean plants to 80 nm and 500 nm PS-MNPs. While the 500 nm PS was held on the leaf epidermis probably due to the large size, the 80 nm was observed in the apoplast and in the cytoplasm having migrated through the stomatal and cuticular pathways [130]. Sun et al. demonstrated the foliar uptake and leaf-to-root translocation of nanoplastics (PSNPs-NH_2_), together with the promotion of uptake of other pollutants by nanoplastics, as PSNPs-NH_2_ promoted phthalate esters (PAE) bioaccumulation in corn leaves and roots. PSNPS-NH_2_ and PAEs damaged the photosynthetic machinery of the plant, significantly inhibiting its growth [117]. The adherence of the large size MNPs to plant surfaces blocks the stomata, while on root surfaces, MNPs alter the shape of root epidermal cells, significantly affecting the uptake of water and nutrients through root hairs [131]. 

MNP accumulation in plants may result in toxicity effects to the plant and other organisms through the food chain. Their presence in food crops, fruits, and vegetables, not only impair the food quality but poses a human health risk [132].

### 3.2. MNP Uptake by Animals

Animals are exposed to MNPs through the air, water, and soil. MNPs are ubiquitous in the food chain and water supplies; hence, these particles are easily taken in through diet and through drinking water (Figure 3), while MNPs in the air are taken in through inhalation. Aquatic animals, especially fish, are exposed to MNPs in water systems and they take in these MNPs through diet and drinking water [124]. Wardlaw et al. showed that the demersal river fish (from the upper Thames River, Ontario), ingested fragments, fibers, and tire wear particles [133]. Justino et al. showed that fish ingest more MNPs in the upper mesopelagic layer than the lower mesopelagic layer, with the fibers of PA, PE, and PET being the most prevalent [134]. Clark et al. demonstrated the uptake and internalization of MNPs in rainbow trout (across the gastrointestinal tract) by tracking using palladium-doped PSNPs (~200 nm). Measurable amounts were determined in fish organs (intestine, liver, gallbladder, kidney, gill, and carcass), with the largest fraction found in the carcass (muscle, bone, and sinew) [135]. This raises health concerns as the muscle, bone, and sinew are the parts taken in a human diet. A pilot study by Van-der Veen et al. revealed the presence of plastic particles in beef and pork (from cows and pigs of Dutch farms, Netherlands) [136]. MPs have also been reported in raw fresh milk [137]. This shows that animals can take up and internalize MNPs. 

MNPs due to their small size can cross biological barriers and accumulate in tissues and organs, inducing cellular and molecular changes, which result in toxic effects. Sendra et al. demonstrated the effect of size (50 nm, 100 nm, and 1 μm) on the internalization and translocation of PSNPs to *Mytilus galloprovincialis* hemocytes, as only 50 nm NPs were detected in the digestive gland and muscle [118]. Li et al. also reported the rapid internalization of PSNPs (50 nm) by murine splenic lymphocytes, which induced reactive oxygen species production, resulting in oxidative stress and lymphocytic structural damage of the mitochondria [138]. The larger specific surface area of NPs also allows them to absorb/adsorb environmental toxins, which are also internalized enhancing the toxicity effects. Yu et al. demonstrated that co-exposure of PSNPs and oxytetracycline (antibiotic), promoted the internalization of oxytetracycline, enhancing the toxicity effects (intestinal damage) in zebrafish [119]. PSNPs were also reported to promote accumulation of microcystin-LR (toxins) in the liver of zebrafish [139]. 

MP can also be transferred up the food chain to humans through diet. Kim et al. demonstrated the trophic transfer by exposing NPs to algae, small crustacean, and fish through trophic transfer experiments. Results indicated that NPs adsorbed to the algal cell wall and were subsequently transferred to higher trophic level organisms through diet, as indicated by their accumulation in fish [140]. Cary et al. demonstrated that ingested PS nanospheres can be translocated to the placenta and fetal tissues of pregnant rats [141], an indication of possible trans-generational NP transfer. Uptake by both animals and humans is concerning considering the toxicity effects (Figure 4) associated with these nanosized particles [142].

### 3.3. MNPs Uptake by Humans

MNPs are taken up by humans through diet, drinking water, dermal contact, and inhalation (Figure 3). The presence of MNPs in food crops, fruits, vegetables, animal products, and drinking water suggests that humans take them in through their diet. Respiratory tract inhalation exposes organisms to atmospheric MNPs, and the major effects of the exposure are felt in the lungs, leading to respiratory diseases [143] including lung cancer. Amato-Lourenço et al. observed MPs (mostly polyethylene and polypropylene) in human lung tissues from autopsies [144] confirming uptake through inhalation. Schwabl et al. detected several microplastics in human stool [145], confirming uptake of the particles from either food or drinking water. Leslie et al. reported the presence of plastic particles in human blood [146] which raises health concerns as in vitro and in vivo studies have demonstrated their toxicity. 

The ability of NPs to build up in organisms over time (bioaccumulation) and increase in concentration as they are passed up the food chain (biomagnification) influence their toxicity levels. Bioaccumulation and biomagnification govern the extent of NPs transport within the organism and food chain. Brandts et al. exposed goldfish (*Carassius auratus*) to 44 nm PSNPs (through water) for 30 days; however, no bioaccumulation was found in the gastrointestinal tract but NPs bioaccumulated in the fish liver and muscle [147]. Bioaccumulation in fish muscle raises concerns of possible biomagnification since fish muscle is consumed by humans. NPs uptake, accumulation, and toxicity have been reported in the food chain. Various primary producers have been reported to accumulate NPs, with toxicity effects reported in microalgae *Chlorella vulgaris* [148], microalgae *Rhodomonas baltica* [149], as examples in aquatic plants, and wheat and lettuce crops [150] as examples in terrestrial plants. The accumulated NPs in primary producers can be transferred to higher trophic levels through diet [140], where they bioaccumulate and display toxicity effects.

## 4. Mechanism of MNP Toxicity

MNPs have displayed several toxicity effects to living organisms, as illustrated in Figure 4. The toxicity effects are mostly due to their small size and high surface area; however, other factors such as surface charge, presence of functional groups, exposure time, MNP concentration, particle shape, and polymer type, to mention a few, also influence their toxicity [151]. Particle size promotes the internalization and accumulation of MNPs in the cells; hence, NPs are the most internalized compared to the relatively larger MPs. However, it should be noted that these MNPs can be found in the GI tract since they can be taken in drinking water or through diet. Additionally, small sizes are associated with increased surface energies and interactions. Surface functional groups and surface charge of MNPs control their interactions (reactivity and stability) and movement [152]. Furthermore, a higher specific surface area to volume ratio results in more interactions with the biomolecules surfaces together with enhanced adsorption ability for other contaminants.

### 4.1. Cytotoxicity of MNPs

Cellular toxicity (cytotoxicity) is greatly influenced by the MNP size [153], the presence of functional groups on MNPs surfaces, and the surface charge (which enables them to interact or attach to other biomolecules in the cell). Small-size MNPs easily cross the cell membrane and accumulate in cells [153]. Li et al. displayed the role played by the particle size and surface charge on the cytotoxicity of PSNPs to murine splenic lymphocytes. Comparatively, the larger (50 nm) PSNPs penetrated into splenic lymphocytes and were taken up more efficiently by the cells than the smaller (20 nm) PSNPs, suggesting that 50 nm was the right size for cellular internalization and transport. Generally, the cytotoxicity of these PSNPs was shown by the reduced cell viability, induction of cell apoptosis, up-regulation of apoptosis-related protein expression, generation of reactive oxygen species, mitochondrial membrane potential alteration, and mitochondrial function impairment. Positively charged PSNPs (20 nm PS-NH_2_-NPs) exerted stronger toxicity than negatively charged (20 nm, PS-SO_3_H-NPs) and uncharged NPs (20 nm PS-NP) [138]. Work by Chen et al. also showed that amino functionalized PSNPs were endocytosised into cells more than the carboxyl functionalized and pristine counterparts, inducing membrane damage as a principal cytotoxicity mechanisms to mouse mononuclear macrophage (RAW264.7) cells [154]. Yuan et al. utilized molecular dynamics simulations and noted that interactions between MNPs (polypropylene, polyvinyl chloride, polylactic acid, polystyrene, and polyethylene terephthalate) and the lipid (dipalmitoylphosphatidylcholine (DPPC)) bilayer were mostly due to van der Waals interactions (instead of electrostatic interactions) and this reduced the thickness of the lipid bilayer, signaling cytotoxicity [155]. Polystyrene MNPs have displayed cytotoxicity and genotoxicity in human lung (A549) cells, with small size and surface-modified (amino-fuctionalised) NPs being easily internalized in cells, displaying a stronger ability to inhibit cell viability [156]. 

Since MNPs exist with other pollutants in the environment, they can adsorb other pollutants due to their large surface area. This promotes the uptake of pollutants into organisms resulting in toxicity from both the MNPs and the pollutants. For example, molecular simulations by Cheng et al. showed that co-exposure of PSNPs and benzo[a]pyrene in aqueous environments encourages adsorption of benzo[a]pyrene onto NPs. This facilitates entry of benzo[a]pyrene into the DPPC bilayer and a combination of the two enhances cytotoxic effects [157]. In vitro studies by Yan et al. using human colon adenocarcinoma cells (Caco-2 cells) confirm the enhancement effect from co-exposure of NPs and other compounds/pollutants, as PSNPs enhanced the cytotoxicity of okadaic acid to Caco-2 cells [158].

### 4.2. Genotoxicity of MNPs

MNPs and chemicals adsorbed on their surface have shown capability to cause damage to genetic information in cells. Exposure to these MNPs or chemical agents consequently results in genomic instabilities and alterations that manifest in the form of diseases such as cancer in humans. Shi et al. evaluated the genotoxicity of PSMPs (2 μm) and PSNPs (80 nm), in A549 human lung cells, using unmodified and also surface-modified (carboxy and amino groups) polystyrene. The NPs showed more genotoxicity than the MPs, showing the effect of particle size, since NPs can be more easily internalized than MPs. Additionally, surface functionalization of PSNPs, especially with amino groups, promoted their internalization in A549 cells [156]. Brandts et al. demonstrated the genotoxicity of NPs to fish. *Carassius auratus*, on exposure to PSNPs (44 nm/100 μg/L) through drinking water for a month, displayed increased erythrocyte nuclear abnormalities, an indication that these NPs entered the cell nucleus and caused DNA damage [147]. 

The genotoxicity of MNPs in the environment can be enhanced by the presence of other co-pollutants (since these MNPs exist as a mixture with other components (natural or foreign/pollutants). Soto-Bielicka et al. showed that co-exposure of tetrabromobisphenol A (25 µM; a flame retardant found in some polymers) and PSNPs (40 nm;10 µg/mL) caused a significant rise in oxidative DNA damage to *Oncorhynchus mykiss* cell lines [159]. Enhanced genotoxicity on co-exposures of MNPs and other pollutants has also been demonstrated by Barguill et al. using PSNPs and arsenic [160].

### 4.3. Immunotoxicity of MNPs

MNPs have been shown to alter the structure or function of the immune system. Generally, when xenobiotics enter cells, phagocytes engulf and digest them. Monocytes/macrophages, in addition to the regulation of chronic inflammatory and immune responses, secrete chemicals (signaling chemicals) that modify cell behavior [161]. MNPs once internalized disturb intracellular signaling pathways, modifying the immune homeostasis and resulting in toxicity (through either immunosuppression or immunostimulation) [161]. For example, Cheng et al. demonstrated that PSNPs trigger significant hepatic immune toxicity and also stimulate steroid hormone biosynthesis in zebrafish larvae [162]. Accumulation of steroid hormones may lead to immune-related diseases. 

MNPs exist in the environment, especially aquatic systems, with other substances and as such they have been shown to enhance the immunotoxicity of other pollutants. For example, Han et al. demonstrated the immunosuppression of thick-shell mussels *Mytilus coruscus* on co-exposure to PSMPs (500 nm; 0.26 mg/L) and antibiotics (oxytetracycline, 270 ng/L, florfenicol, 42 ng/L: and sulfamethoxazole, 140 ng/L) [163]. 

### 4.4. Oxidative Stress and Inflammation Induced by MNPs

Oxidative stress manifests when there is an imbalance between oxidants and antioxidants which disturbs the redox signaling system, consequently inducing cellular damage. Inflammation is the immune system’s reaction by biological tissues to injurious stimuli. Inflammation and oxidative stress are connected as one could induce the other. MNPs have been shown to cause oxidative damage to *Cyprinus carpio* gills [164]. Tang et al. demonstrated that PSNPs can stimulate oxidative stress and trigger the mitogen-activated protein kinase (MAPK) pathway, resulting in inflammation and necroptosis in mice spleen and RAW264.7 cell line [165]. A similar observation was reported by Chen et al; however, functionalization of the PSNPs with amino and carboxyl groups enhanced the inflammatory potential [166]. Tang et al. also observed that co-exposure of PSNPs with lipopolysaccharide enhanced inflammatory damage [165]. While functionalization improves the internalization and interaction of MNPs with cells, MNPs’ co-exposure with other pollutants not only promote the internalization of these pollutants (which are mostly adsorbed on these MNPs) but enhance their toxicity. Woo et al. demonstrated that mitochondrial damage induced by polypropylene NPs causes lung inflammation using the p38 phosphorylation-mediated nuclear factor kappa (BNF-κB) pathway [167]. Trans-generational PSNPs toxicity was demonstrated by Huang et al., as maternal exposure during gestation and lactation induced an inflammatory response in the liver of mouse offspring [168]. Inflammation is associated with cancer, heart disease, inflammatory bowel disease, rheumatoid arthritis, and other ailments.

### 4.5. Gastrointestinal Alterations and Membrane Injury due to MNPs

MNPs have been reported in the gastrointestinal tract. Their presence in human and animal feces is evidence of dietary uptake or uptake through drinking water [169]. MNPs have been detected in beer, wines, rice, table salts, and honey, and in fruits and vegetables (apples, broccoli, and carrots) having been taken up either through their root systems or foliar uptake. Ingestion of MNPs has been shown to cause intestinal epithelial cell injury and induce alterations of the intestinal microflora and intestinal barrier function [170]. Hao et al. demonstrated that intestines can greatly accumulate MNPs (as both sizes of nanoplastics studied (86 and 185 nm) were detected) and severe intestinal mucosal layer damage was a result of the smaller rather than larger- sized MNPs. Larger MNPs induced superior impact on microbiota composition [171]. Wang et al. reported the structural damage of vascular endothelial cells in mice on exposure to PSNPs (PS-NH_2_ and PS-COOH) [172].

### 4.6. Neurotoxicity of MNPs

Neurons transmit and process signals in the brain and other nervous system parts; however, toxins can alter or disrupt these activities and at times destroy them. MNPs have been reported to accumulate in the brain [173,174,175]. In vivo studies by Shan et al. demonstrated that PSNPs can induce the permeability of the blood–brain barrier, allowing their accumulation in the brain. Additionally, these MNPs were observed in the microglia, and were reported to induce microglia activation causing neuron damage in the mouse brain [173]. MNPs have been shown to inhibit the acetylcholinesterase activity, changing neurotransmitter levels, resulting in behavioral changes [174]. Zhou et al. demonstrated that PS-NPs (100 and 500 nm) can cross the embryo`s chorionic pores, translocate to other tissues including the brain causing neurotoxicity (as displayed by less larval neurons, axonal defects in motor neurons, and neuronal apoptosis). Their work revealed that the mechanism of neuronal injury by MNPs might be through inducing abnormalities in development-related and apoptotic genes. Additionally, PS-NPs altered synaptic signaling and contribute to the development of neurotoxicity by inducing abnormalities in the neurotransmitter system [175].

### 4.7. Carcinogenicity of MNPs

While plastic monomers (such as styrene or vinyl chloride, to mention a few) are toxic and carcinogenic, (with some plastics leaching hazardous chemicals including additives,) their micro/nanosized counterparts are potentially more toxic and a threat to health. Barguilla et al. demonstrated the potential carcinogenic risk of PSNPs, as long-term exposure to MNPs modified functional and molecular characteristics linked to the carcinogenic process [176]. Sulukan et al. using zebra fish demonstrated that parental exposure to MNPs may continue to interrupt many cancer-associated processes even in the following generations [177]. With the rising cancer burden including childhood cancer [178,179,180,181], environmental pollutants [182,183] including MNPs could be contributors towards the cancer surge. Due to their large surface area to volume ratio, these MNPs adsorb chemicals, some of which are persistent organic pollutants and/or endocrine-disrupting chemicals (EDCs) [184]. EDCs interfere with the hormone function of organisms including animals or humans. These chemicals, found in the plastics utilized daily, are reported to contribute to cancer development [183,185]. For example, Deb et al. demonstrated that the EDC, bisphenol-A prompts the homeobox gene, HOXB9 expression in vitro (cultured MCF7 human breast cancer cells) and in vivo (in mammary glands of ovariectomized rats) [186]. Altered HOXB9 expression is linked to a range of cancers, together with breast, colorectal, and lung cancer [187]. Maternal exposure to MNPs during gestation and lactation periods has been shown to be responsible for inflammatory response and glucometabolic disorder in the liver of mouse offspring [168].

Other toxicity effects such as growth inhibition, behavior alteration in animals, hepatotoxicity, and hematotoxicity to mention a few have been reported [188]. These toxic effects of MNPs show the need for elimination of plastic pollution in the environment to avoid human, plant, and animal health effects. There is a need for governments to be involved to speed up the total removal of plastic waste. 

## 5. Future Perspectives

The degradation of plastic waste is a global concern because MNPs pose threats to aquatic flora and fauna, including birds and humans. This work highlighted the transformation of plastic waste through degradation to forms that cause harm to the environment. With the increased plastic production and use, high levels of MNPs are continuously being released into the environment causing toxicity effects to all living organisms. Consequently, drastic measures to minimize plastic waste have to be taken. Since MNPs can be transported over long distances even to remote regions through air and water, this makes them a global problem requiring international collaborations.

Joint international efforts are required in reducing plastic waste, which should start with research methods for monitoring, controlling, and evaluating microplastic transformation and degradation in various environments. 

Policies that promote plastic use reduction should be formulated and implemented. There are already global initiatives to reduce plastic waste; however, there are disparities between the global North and South in these efforts. For example, currently, there are bans on single-use plastics [189,190], and if implemented this could significantly reduce the levels of plastic waste in the environment, together with their toxicity effects. Unfortunately, some low and middle-income countries are struggling to implement plastic bans [190]. This is exacerbated by the porous borders that characterize low-income countries, as such implementation of policies and strict regulations will require multistakeholder and regional governments’ cooperation and action against single-use plastics.

Waste management policies also need to be implemented both at the regional and international levels, with low and middle-income countries partnering with high-income countries where capacity is lacking to develop innovative technologies in plastic waste management. For example, a starting point for low and middle-income countries could be in waste sorting and collection, which can minimize waste dumping in water systems, along roads, and accessible spaces. In contrast, waste traps in drainages can capture these before release to water systems. Consumer education is another example that can be implemented. When consumers are educated about the impact of this waste and trained on how to manage it, they will participate in all plastic reduction measures, including the stoppage of littering. Alternatively, repurposing plastic waste into value-added products could be an opportunity for enterprise development in low and middle-income countries, which are plagued with joblessness and poverty. For this to be realized, there will be a need for a change of mindset and environmental education.

## 6. Conclusions

This review highlighted the increased production and use of plastics, and poor plastic waste management, which consequently results in the presence or occurrence of MNPs everywhere (atmosphere, hydrosphere, and lithosphere). The uptake and internalization of MNPs by plants, animals, and humans are discussed, together with the resulting toxicity effects. 

MNPs, though very small, have a huge negative impact in the environment. Measures such as landfilling and incineration only change the pollutant from one form to the other. Unfortunately, the plastic form produced is at times not visible to the eye and causes more harm to the environment than the pristine form. 

There is a need for total irradiation of plastic waste to avoid the formation of large quantities of toxic MNPs. Effective plastic waste management should start with regular waste collection for transformation or value addition through circular plastic economy initiatives (Reduce, Reuse, Recycle) [191]. It is important to minimize plastic pollution and this is possible if communities are educated in order for them to be empowered and involved in plastic pollution reduction. Alternatively, eco-friendly materials should be utilized in place of plastics.

## Figures and Tables

**Figure 1 ijerph-20-06667-f001:**
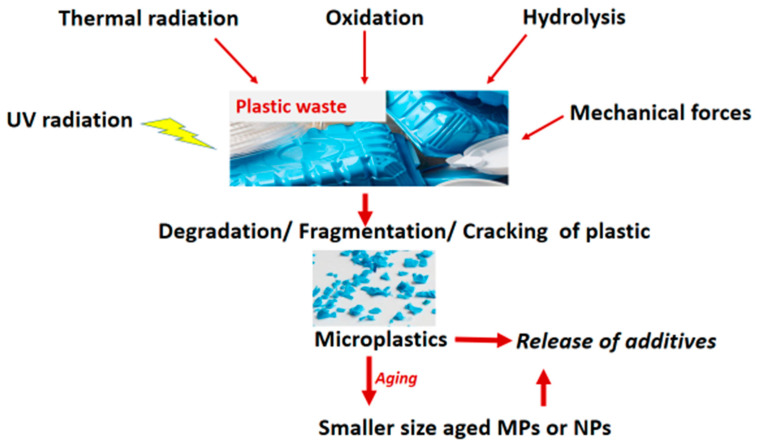
Degradation of plastics to MNPs and associated contaminants.

**Figure 2 ijerph-20-06667-f002:**
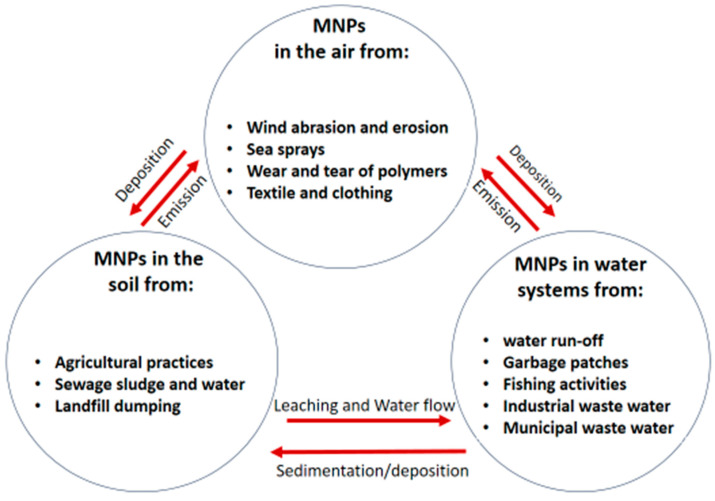
Entrance and occurrence of MNPs in the soil, water, and air.

**Figure 3 ijerph-20-06667-f003:**
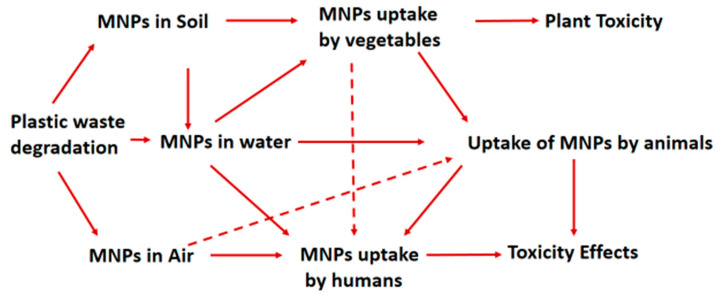
Illustration of the transfer of MNPs in soil, water, and air to plants, animals, and humans, resulting in toxic effects.

**Figure 4 ijerph-20-06667-f004:**
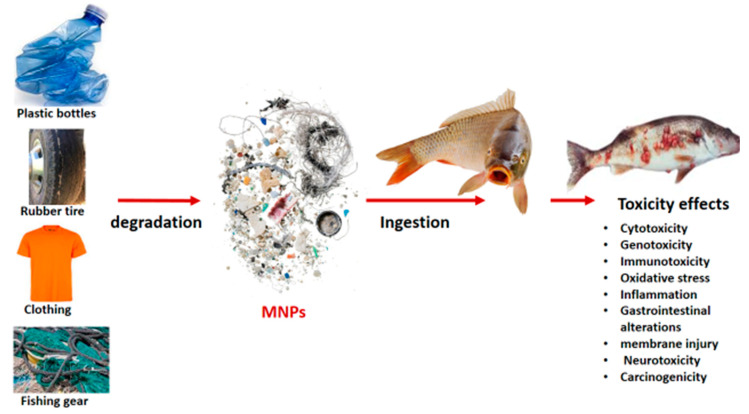
Illustration of the degradation of plastic and polymeric materials, MNPs formation, uptake by living organisms (represented by fish as an example), and the resulting toxicity effects.

## Data Availability

New data were not generated for this study.

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
