# Peer review of "Plastics and Micro/Nano-Plastics (MNPs) in the Environment: Occurrence, Impact, and Toxicity"

_ijerph, 2023, doi:10.3390/ijerph20176667_

Round 1

Reviewer 1 Report

Consider the following comments below for the improvement of your manuscript:

1)    In page 1, line 28-31; kindly include citation at the end of that sentence to back up the data you claimed.

2)    In page 2, line 53-54: the specific types of toxins emitted and their potential effects on human health need to be mentioned. Instead, add this Additionally, incomplete incineration of plastic often leads to the release of numerous toxins, including volatile organic compounds, polybrominated dibenzo-p-dioxins and furans, heavy metals, polycyclic aromatic hydrocarbons, polychlorinated biphenyls, and various other gaseous emissions into the environment [x]. These toxins have direct and severe health consequences, which include an increased risk of cancer and respiratory diseases, neurological disorders, damage to the immune and nervous systems, and cardiovascular diseases [xx].” Kindly include this reference to back up the data you claimed.  10.1016/j.enconman.2022.115758.

3)    Please ensure that the entire manuscript adheres to the proper heading format. There are inconsistencies with some headings, particularly on page 4, lines 144, 159, etc.

4)    Please, include more figures and tables in your review article, as they are essential elements that enhance the clarity, readability, and impact of the content. They provide a visual representation of data, help summarize key findings, and strengthen the overall argument of the review.

5)    What is the practical implication of this study? Also, discuss and explain the appropriate policies should be based on your research findingsIt is strongly recommended to add this as a sub-section, before the conclusion.

Reviewer 2 Report

The Review entitled “Plastics and micro/nano-plastics (MNPs) in the environment: occurrence, impact, and toxicity” aim to conduct a traditional review of the current state of the art regarding microplastics (MPs) definition and characterization. More specifically, highlights the entry and occurrence of primary and secondary MNPs in the environment (soil, water and air), together with their aging. Furthermore, the uptake and internalization of MNPs by plants, animals and humans are discussed, together with their toxicity effects.

The authors address an important extremely current topic. Plastic pollution in various forms has emerged as the most severe environmental threat.

The level of plastic debris, mainly as microplastics, in the environment is reaching unprecedented levels and there is worldwide concern about their adverse effects on both living organisms (including humans) and the environment.

It is also vital to ensure that publishes articles that raises public awareness to convey specific messages related to threats and potential opportunities to reduce impacts from microplastics.

The completeness of the topic can be improved:

I strongly suggest including in the sections:

3 Uptake and internalisation of MNPs by living organisms

3.2 MNPs uptake by animals

Guerrera, M.C.; Aragona, M.; Porcino, C.; Fazio, F.; Laurà, R.; Levanti, M.; Montalbano, G.; Germanà, G.; Abbate, F.; Germanà, A. Micro and Nano Plastics Distribution in Fish as Model Organisms: Histopathology, Blood Response and Bioaccumulation in Different Organs. Appl. Sci. 2021, 11, 5768. doi.org/10.3390/app11135768, since this is a complete review that reviewed 219 until 2021.

and

3.3. . MNPs uptake by humans

Yee, M.S.-L.; Hii, L.-W.; Looi, C.K.; Lim, W.-M.; Wong, S.-F.; Kok, Y.-Y.; Tan, B.-K.; Wong, C.-Y.; Leong, C.-O. Impact of Microplastics and Nanoplastics on Human Health. Nanomaterials 2021, 11, 496. https:// doi.org/10.3390/nano11020496

The review is clear, comprehensive and of relevance to the field

Although similar reviews have been published on the topic, this current review still relevant and of interest to the scientific community

The paper can in principle be accepted after revision based on the reviewer comments.

Minor editing of English language required

Reviewer 3 Report

I would like to acknowledge the effort and diligence you've put into your study on MNPS in various environments. I have a few observations and suggestions that might enhance the clarity and completeness of your work:

1. In line 133, I noticed that the content below the title seems a bit sparse. It might be beneficial to provide a more detailed discussion or elaboration in this section to offer readers a comprehensive understanding.

2. Regarding line 144, I found the use of "iiii" for subsection designation a bit unconventional. Could you consider revising it to a more commonly accepted format?

3. I'd recommend incorporating findings from a recent study on airborne microplastics, which could enrich your current discussion:

   Xiao, S., Cui, Y., Brahney, J., Mahowald, N., & Li, Q. (2023). Long-distance atmospheric transport of microplastic fibers depends on their shapes.

Your research is pivotal in this domain, and these suggestions are aimed at ensuring that it shines in its best light. I trust you will find these comments constructive and valuable.

Reviewer 4 Report

The study has the potential to attract the attention of readers in this field. However, there are some problems with some parts of this study. These problems should be corrected and these parts of the study should be developed in line with the recommendations.

Some of these are as follows.

-If this study is a review study, it is expected that the subject will be handled comprehensively from many papers. The content of many subheadings between page 4 and page 7 should be improved by adding new examples.

-Some of the references in the work do not have reference numbers, and some are not even on the reference list.

-The content of some titles remained rather weak. For example, under the heading MNPs uptake by animals, fish that are highly affected by microplastics are hardly mentioned in this section (Almost no study about microplastic uptake by fish species in this section). This is one of the biggest shortcomings. Moreover, A similar situation applies to most of the other headings, and these sections should be developed to include the heading.

-Species and genus names should be written in italics.

-There are some problems in the sub-titles of the study, some parts are only nanoplastic and some are formed in a micro/nanoplastic way. These title names should also be compatible.

-Other suggestions and corrections are in the attached file.

Round 2

Reviewer 1 Report

The reviewer's comments have been comprehensively addressed. Therefore, the manuscript is deemed suitable for publication.

Reviewer 4 Report

After the review of the MS, it was determined that the suggested arrangements were made by the author